# MULTIMODAL QUESTION ANSWERING FOR UNIFIED INFORMATION EXTRACTION

## ABSTRACT

Multimodal information extraction (MIE) aims to extract structured information from unstructured multimedia content. Due to the diversity of tasks and settings, most current MIE models are task-specific and data-intensive, which limits their generalization to real-world scenarios with diverse task requirements and limited labeled data. To address these issues, we propose a novel multimodal question answering (MQA) framework to unify three MIE tasks by reformulating them into a unified span extraction and multi-choice QA pipeline. Extensive experiments on six datasets show that: 1) Our MQA framework consistently and significantly improves the performances of various off-the-shelf large multimodal models (LMM) on MIE tasks, compared to vanilla prompting. 2) In the zero-shot setting, MQA outperforms previous state-of-the-art baselines by a large margin. In addition, the effectiveness of our framework can successfully transfer to the few-shot setting, enhancing LMMs on a scale of 10B parameters to be competitive or outperform much larger language models such as ChatGPT and GPT-4. Our MQA framework can serve as a general principle of utilizing LMMs to better solve MIE and potentially other downstream multimodal tasks.[1]

## 1 INTRODUCTION

Multimodal information extraction (MIE) aims to extract structured information from unstructured multimedia sources, which has drawn increasing attention as social media platforms are flooded with multimedia contents. Typically, with text and images as input, MIE can be categorized into three specific sub-tasks including multimodal named entity recognition (MNER; Sun et al. (2021)), multimodal relation extraction (MRE; Chen et al. (2022b)), and multimodal event detection (MED; Li et al. (2020a)), each with its own output format, as depicted in Figure 1a.

In addition to varying task formats, another major challenge of MIE lies in the diversity of task settings. Taking MED as an example, event triggers can be text-only, with images serving as supplementary materials (Zhang et al., 2017). Conversely, images can be used as the main component, while texts play an auxiliary role (Li et al., 2022). Finally, event triggers can also show up in both text and images simultaneously (Li et al., 2020a).

Given the diversity of tasks and settings, current works design task-specific models to address each challenge separately. Moreover, these methods require a large amount of labeled data for training, which may not always be available in many real-world scenarios. As a result, such a *"one task, one model"* paradigm is not only data-intensive and time-consuming, but also limits model's ability of generalizing to new tasks or datasets. This becomes a major bottleneck in situations with limited resources or where fast adaptation is necessary.

To address these issues, in this work, we propose a novel multimodal question answering (MQA) framework to unify all aforementioned MIE tasks under diverse settings, as briefly illustrated in Figure 1b. Building upon off-the-shelf large multimodal models (LMMs), MQA framework can uniformly address these tasks in a zero-shot fashion, thus keeping its generalization capabilities. Specifically, we decompose all MIE tasks into cascaded atomic task forms, namely span extraction (optional) and classification tasks. Furthermore, we design multi-choice question answering

---

[1]Code will be released upon acceptance.

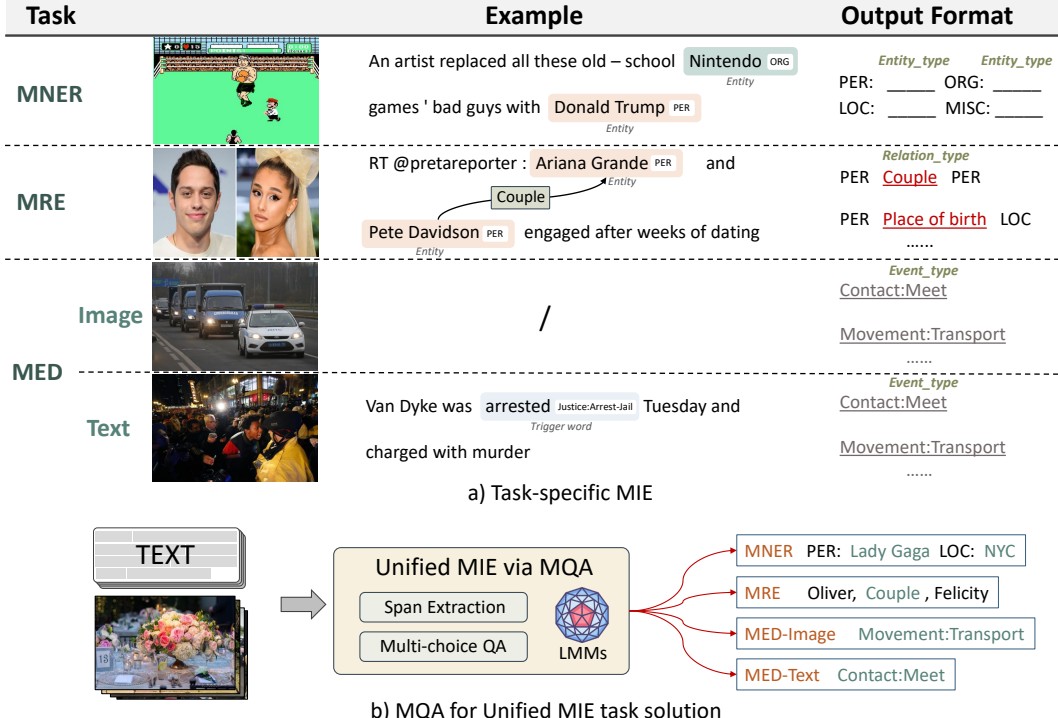

Figure 1: a) Illustration of MIE tasks, including input examples and output format. b) Our proposed MQA framework for LMMs to unify various MIE tasks.

(QA) to elicit the classification abilities of various LMMs (Li et al., 2023b; Dai et al., 2023) by aligning (Zhang et al., 2023a) classification tasks with pre-trained task forms (QA) of LMMs.

Comprehensive experiments on six LMMs across six datasets from three MIE tasks show that the MQA framework outperforms vanilla MIE prompting strategy by a considerable margin. Moreover, without any training data (zero-shot), MQA achieves significantly better results than previous state-of-the-art (SOTA) zero-shot and few-shot methods tailored for each task, and much larger language models including ChatGPT (OpenAI, 2022) and GPT-4 (OpenAI, 2023) on most of the datasets. Such empirical results demonstrate the effectiveness of MQA framework on MIE tasks and its general applicability on LMMs, and underscore the potential of task reformulation for better adapting LMMs to other downstream multimodal tasks. To summarize, our contributions are as follows:

• To address the task diversity and generalization issue, we propose a novel MQA framework to uniformly decompose three MIE tasks into cascaded atomic tasks: span extraction and classification. Also, we design multi-choice QA reformulation to elicit the classification abilities of various LMMs. To the best of our knowledge, we make the first attempt to unify MNER, MRE, and MED tasks.

• Extensive experiments of six LMMs on six datasets from three MIE tasks show that our MQA framework significantly outperforms vanilla prompting strategies. With MQA, LMMs surpasses previous low-resource SOTA and much larger language models including ChatGPT and GPT-4 on most datasets. Further in-depth analysis shows strong robustness of our MQA framework and consistent effectiveness in the few-shot setting.

• Finally, our work provides discussions and insights on recent LMMs from a MIE perspective. Also, our study indicates the potential of task reformulation, serving as a promising general principle for better adopting LMMs to other downstream multimodal tasks.

## 2    RELATED WORK

**Multimodal Information Extraction.** MIE tasks mainly include multimodal named entity recognition (MNER), multimodal relation extraction (MRE), and multimodal event detection (MED) with

images and texts as input. **MNER** (Moon et al., 2018; Lu et al., 2018; Arshad et al., 2019; Yu et al., 2020; Sun et al., 2021; Wang et al., 2022; Chen et al., 2023) aims to identify mentions and classify them into pre-defined categories from texts, using images as additional information. **MRE** (Zheng et al., 2021b;a; Chen et al., 2022b) aims to infer the relationship between head and tail entities based on the given image. **MED** (Zhang et al., 2017; Li et al., 2020a; 2022; Liu et al., 2022) is a task of identifying event triggers and their corresponding event types. In context of text, event triggers typically refer to specific words or phrases that signify the occurrence of an event. According to Li et al. (2020a), event triggers can be spans in the given texts or be the given images per se, showing the challenging nature of MED task.

**Low-Resource Information Extraction.** To address the challenge of data scarcity in the domain of multimodal information extraction, several methods are proposed to leverage transfer learning and data augmentation techniques. A number of works (Yu et al., 2020; Zhang et al., 2021; Lu et al., 2022a; Chen et al., 2023) train the models with one dataset from the same domain and evaluate them on another (e.g., from Twitter15 to Twitter17). However, such transfer learning still requires a sufficient alignment between the source and target domains, which is crucial for attaining optimal performance. PGIM (Li et al., 2023a) utilizes ChatGPT (OpenAI, 2022) for additional knowledge generation, thus augmenting the downstream models for MNER tasks in order to achieve better few-shot performances. Chen et al. (2022a) propose MKGFormer to bridge the modality gap between text and image inside Transformers (Vaswani et al., 2017), achieving decent performances in low-resource scenarios of MRE and MNER. Li et al. (2020a) adopt annotated uni-modal corpora to separately train textual and visual event detection models, and bridge their modality gap with an image-caption. In a contrastive learning fashion, Li et al. (2022) pre-train a vision-language model by connecting events and arguments across different modalities. These weak supervision methods achieve impressive MED results in a zero-shot setting.

**Large Multimodal Models.** Pioneering multimodal models (Lu et al., 2019; Tan & Bansal, 2019; Zhou et al., 2020; Chen et al., 2020; Li et al., 2020b; 2022) are typically pre-trained on vision and language data to establish connections between modalities. As evidenced by the remarkable capabilities demonstrated by large language models (Brown et al., 2020; Chowdhery et al., 2022; Touvron et al., 2023a;b; Chung et al., 2022) especially when increasing the model size and training tokens, recent multimodal works focus on equipping off-the-shelf language models with visual abilities via lightweight fine-tuning (Li et al., 2023b; Dai et al., 2023; Liu et al., 2023; Zhang et al., 2023b; Zhu et al., 2023; Ye et al., 2023). Meanwhile, instruction tuning (Lou et al., 2023; Ouyang et al., 2022) empowers these models to follow instructions, thus enabling them to generalize to unseen tasks. However, Zhang et al. (2023a) shows that instruction-tuned models fail to deliver decent IE results and highlight the potential benefits of reformulating IE as a QA task. Our work mainly builds on these prior efforts while we extend it into multimodal scenarios and unify three MIE tasks.

# 3 MULTIMODAL QUESTION ANSWERING FRAMEWORK

## 3.1 PRELIMINARY

We provide formal definitions for MNER, MRE, and MED tasks as follows.

**Multimodal Named Entity Recognition.** Given a sentence $T$ and an associated image $I$, the task of MNER requires models to identify disjoint continuous spans within the sentence $T$ and subsequently classify each span into one of the pre-defined entity types, such as person and location. The image $I$ mainly serves as an additional clue to enhance the extraction of textual entities.

**Multimodal Relation Extraction.** MRE aims to infer the relationship between two entities based on both a sentence and its paired image. Concretely, each instance of a relation contains a sentence $T$ and an associated image $I$, along with a head entity $E_h$ and a tail entity $E_t$ within $T$. Given a relation example $(S, I, E_h, E_t)$, models are required to identify the relation between $E_h$ and $E_t$ expressed in $T$ from a set of pre-defined relation types with the auxiliary information from image $I$.

**Multimodal Event Detection.** Based on the modality (image or text) in which the event trigger is detected, MED can be further categorized into multimodal image-centric event extraction (MIED) and multimodal text-centric event extraction (MTED). MIED classifies the given image $I$ into event

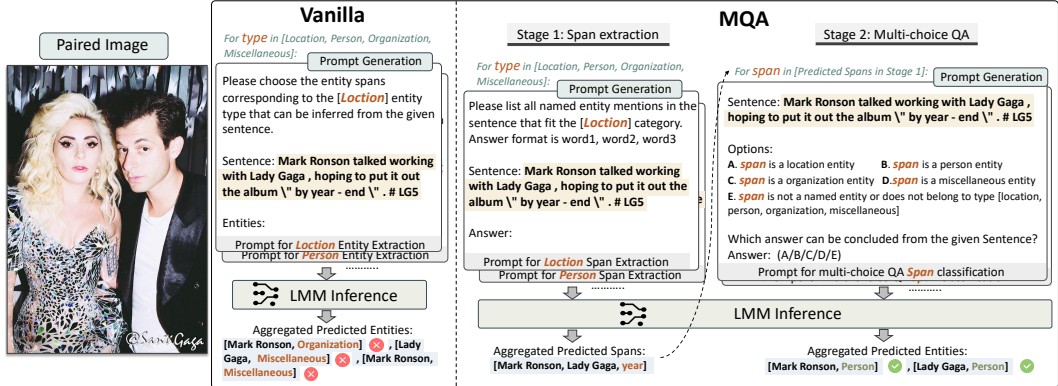

Figure 2: This figure compares the vanilla method and our proposed MQA framework for MIE tasks, using the MNER task as an illustrative example. With the vanilla prompting strategy (Gutiérrez et al., 2022), LMMs directly identify entities. This complicated and error-prone process may lead to inferior results (e.g., one same span can be classified as two different entity types). In contrast, our MQA framework decomposes a MIE task into two cascaded phases: span extraction and multi-choice QA. Spans are extracted as candidates for later multi-choice QA. Each candidate span can be classified into pre-defined categories and one additional *none-of-the-above* option (E) to discard false positives from the span extraction stage.

types. Conversely, similarly to MNER, with image $I$ serving as an auxiliary cue, MTED extracts a set of event mentions $e$ from a given sentence $T$ and categorizes them into event types separately.

## 3.2 TASK DECOMPOSITION

Both MNER and MED can be decomposed into span extraction and span/image classification tasks, and MRE can be regarded as sentence classification conditioned on entities and images. Therefore, we unify these three MIE tasks as optional span extraction and classification tasks.

**Span Extraction.** Given a sentence with $n$ tokens $T = \{t_1, ..., t_n\}$, models are tasked with identifying a set of non-overlapping spans. (e.g., $s = \{t_i, ..., t_j\}$ where $1 \leq i$ and $j \leq n$). For MNER and MTED, these identified spans subsequently serve as candidates for later classification to determine the exact entity/event types individually.

**Multi-choice QA.** Given a span/image/sentence, LMMs need to classify it into pre-defined categories corresponding to the specific task. By providing appropriate instructions and label space in the prompt, LMMs can perform these classification tasks uniformly by generating label names (Gutiérrez et al., 2022). In particular, for MNER and MED, we combine *each* extracted span $s$ with the original sentence $T$ to perform the classification task, with the image $I$ as a visual clue.

## 3.3 MULTI-CHOICE QUESTION ANSWERING

To standardize all classification tasks, we reformulate them as multi-choice QA tasks. As illustrated in the second stage of the MQA framework in Figure 2, we provide LMMs with question answering instructions and inference examples with each possible class as an answer option, enabling LMMs to perform classification by generating an answer index. This empowers the LMMs to harness their extensive VQA abilities gained from either multimodal pretraining or visual instruction tuning to enhance classification performance, as evidenced by Zhang et al. (2023a). Furthermore, by adopting the MQA framework, we streamline the output format diversity across various tasks and alleviate the decoding complexity, resulting in a more unified and effective approach.

## 3.4 ANSWER OPTION CONSTRUCTION

Inspired by prior works (Zhang et al., 2023a; Ma et al., 2023), we represent each class as an answer option. Specifically, since the extracted span candidates may not belong to any pre-defined meaningful classes in MNER and MED, we design an irrelevant class called *none-of-the-above* (NOTA), as

exemplified by option E in the multi-choice QA prompt template depicted in Figure 2. Subsequently, we remove spans that are categorized as irrelevant by LMMs to mitigate the potential occurrence of false positives from the span extraction step before multi-choice QA stage.

## 4 EXPERIMENT SETUP

### 4.1 DATASETS

We evaluate our methods on MNER, MRE, and MED tasks, covering six datasets. **For MNER**, we consider (1) Twitter15 (Zhang et al., 2018) and (2) Twitter17 (Lu et al., 2018), both of which are constructed from social media content and have four entity types including person, location, organization, and miscellaneous. We report micro-averaged F1 following previous works (Sun et al., 2021; Chen et al., 2023). **For MRE**, we adopt (3) MRE-V1 (Zheng et al., 2021b) and (4) MRE-V2 (Zheng et al., 2021a). Upon Twitter15 (Zhang et al., 2018), Twitter17 (Lu et al., 2018), and crawled Twitter data, these two MNER datasets are manually annotated with the relationship between two textual entities based on the visual evidence shown in the paired image. Following Chen et al. (2022a), we use micro-averaged F1 with *None-of-the-Above* (NOTA) relation excluded for fair comparison in low-resource setting. **For MED**, we consider (5) image-centric and (6) text-centric split of $M^2E^2$ (Li et al., 2020a). Following prior work (Li et al., 2020a; 2022), we report micro-averaged F1 without NOTA event type in image-centric setting and micro F1 in text-centric setting.

### 4.2 LARGE MULTIMODAL MODELS

To comprehensively evaluate the MQA framework on various LMMs, we consider two series of LMMs, encompassing a total of six models. Concretely, BLIP-2 (Li et al., 2023b) with various-sized FLAN-T5 (Chung et al., 2022) as the language component, and InstructBLIP (Dai et al., 2023) with different-sized FLAN-T5 and Vicuna (Chiang et al., 2023) as language backbones. Both series of models are trained based on a frozen image encoder and a frozen LLM. BLIP-2 (Li et al., 2023b) incorporates a Q-Former designed to extract visual features, serving as soft prompts for frozen large language models to facilitate text generation. Follow-up work, InstructBLIP (Dai et al., 2023), further enhances BLIP-2 by performing instruction tuning on various vision-language tasks and modifying the Q-Former module to extract instruction-aware visual features.

### 4.3 IMPLEMENTATION DETAILS

In our prompt engineering efforts, we explore various prompt formats and task instructions for both vanilla and MQA methods, utilizing BLIP-2 (Li et al., 2023b) on a subset of 400 examples from each task's validation set. These optimized task instructions and prompt formats are then employed for all the models in subsequent experiments. Detailed information on the prompt formats for vanilla and MQA are available in Appendix A.1. In addition, we design experiments for robustness assessment and few-shot fine-tuning to further validate the superiority of our method over the vanilla approach.

#### 4.3.1 VANILLA

With the vanilla approach, models generate verbalized answers directly (Gutiérrez et al., 2022). For MNER task, we instruct models to extract relevant spans corresponding to each entity type. In the case of MRE, following Zhang et al. (2023a), we adopt entity type constraints to narrow down the label space. Given the sentence and two entities, models are required to generate the relation name from the valid relation set, based on the relationship between two entities expressed in the sentence. For MIED task, models are tasked with categorizing the provided image into a pre-defined event type by generating the corresponding event name. Similar to MNER, for the MTED task, we specify the event type and instruct the model to extract words related to the event.

#### 4.3.2 MQA FRAMEWORK

We break down and unify the MIE tasks into span extraction and span/sentence/image classification. Span extraction can be regarded as optional depending on the specifics of the given task. By

conducting multiple-choice QA tasks, LMMs are equipped to generate an answer index for classification, rather than a complete label name. In practice, we employ two-stage prompts to deconstruct the initial MIE tasks, detailed in Table 1.

**MNER/MTED.** These two tasks can be decomposed into a two-stage process, encompassing span extraction and multi-choice QA. As for the MNER task, the initial stage of MQA focuses on identifying entity spans. This stage is designed to allow the LMMs to generate a considerable number of entity candidates, thereby permitting a certain level of false positive entities in a strategic trade-off. In the second stage, we apply a multiple-choice QA to categorize each extracted span into a predetermined set, while any false positive entities can be classified as NOTA and consequently eliminated. Similarly, for MTED, LMMs are first prompted to extract text spans, and then a multi-choice QA is utilized to categorize the event associated with these spans more accurately.

Table 1: Details of task decomposition.

| MIE Task | Stage 1 | Stage 2 |
|---|---|---|
| MNER/MTED | Span Extraction | Multi-choice QA |
| MRE/MIED | Multi-choice QA | - |

**MIED/MRE.** These two MIE tasks are relatively straightforward, each encompassing a single-stage classification task, and we design appropriate prompts for both tasks respectively. In MIED, the model is prompted to identify the correct event type based on the provided image. For MRE, we specify the head and tail entities with their respective entity types and provide both the image and the entire sentence, requiring the model to select from a pre-defined set of relations.

## 4.4 BASELINES

We adopt several baselines to benchmark our proposed MQA framework. These baselines fall into two main groups: (1) prior SOTA baselines and (2) the most powerful larger language models to date. For prior low-resource SOTA methods, we consider PGIM (Li et al., 2023a) for MNER, which attains its performance through 50-shot few-shot fine-tuning. We adopt MKGformer (Chen et al., 2022a) for MRE, which utilizes 40-shot fine-tuning. For two MED tasks, we adapt the SOTA zero-shot model WASE (Li et al., 2020a) as our baseline.

We also consider the most advanced large-scale language models available to date, exemplified by ChatGPT (OpenAI, 2022) and GPT-4 (OpenAI, 2023). Similar to the prompting engineering on open-source LMMs, we meticulously craft and evaluate prompts for these two language models on the validation set of all datasets except the MIED task, which can not be solved by text-only models. Please refer to Table 18 in Appendix for detailed prompts.

## 4.5 FEW-SHOT FINE-TUNING

To assess the effectiveness of MQA framework in the few-shot setting, following Li et al. (2023a), we conduct a series of 50-shot few-shot fine-tuning experiments across all six datasets.

**Few-shot Data Sampling Strategy.** To ensure the distribution consistency between the subset and full set, we initially calculate the proportion of each category within the training set. Subsequently, we determine each category's sample numbers among the 50 samples. While ensuring at least one sample for each category, we conduct random sampling from the dataset following the ascending order of sample numbers per category. As for MNER and MRE tasks, we follow the original data split. In the case of the MIED and MTED tasks, where specific training splits are not provided, we employ the aforementioned method, sampling 50 training samples from the entire dataset as the training set, 200 samples as the validation set, with the remaining serving as the test set.

**Training Hyperparameters.** We perform few-shot fine-tuning experiments based on BLIP-2 with Flan-T5 XXL due to its exceptional performance in the zero-shot setting. The fine-tuning process is carried out on a single NVIDIA A100 GPU with 80G of memory. Specifically, for the MNER, MRE, and MTED datasets, we freeze only the Flan-T5 language component while fine-tuning other parts. For the MIED dataset, we fine-tune Qformer with both the LLM and vision encoder frozen. We employ the Adam optimizer with a learning rate of 2e-5 and execute the training over 10 epochs with a batch size of 8. We select the model checkpoint with the highest micro-F1 score on the validation set for subsequent evaluation on the test set.

Table 2: Experimental results on six MIE datasets (%). The prior SOTA results for each task are achieved by different models, rather than a single unified model. In this context, we use †, ‡, and * to denote these models, where † represents the PGIM (Li et al., 2023a), ‡ symbolizes the MKGformer (Chen et al., 2022a), and * denotes the WASE (Li et al., 2020a). Moreover, MQA** represents the results of multi-choice QA with ground truth spans from the span extraction stage, serving as a performance upper bound. We highlight the best results in **bold** and mark the F1 score improvements of our MQA framework over vanilla method in green.

| Methods | | MNER | | MRE | | MED | | Average |
|---|---|---|---|---|---|---|---|---|
| | | Twitter15 | Twitter17 | MNRE-V1 | MNRE-V2 | Image | Text | |
| *Baselines* | | | | | | | | |
| Prior SOTA (Zero-shot) | | - | - | - | - | 49.9* | 50.6* | - |
| Prior SOTA (Few-shot) | | 52.2 † | 50.7 † | - | 40.9 ‡ | - | - | - |
| ChatGPT | | 39.7 | 45.2 | 41.2 | 48.3 | - | 13.9 | - |
| GPT-4 | | 42.3 | 54.7 | 61.7 | 61.3 | - | 32.5 | - |
| *BLIP-2 series (Zero-shot)* | | | | | | | | |
| Flan-T5 XL | Vanilla | 22.2 | 21.9 | 40.8 | 41.6 | 15.3 | 13.9 | 26.0 |
| | MQA | 43.8 (+21.6) | 56.8 (+34.9) | 51.7 (+10.9) | 56.5 (+14.9) | 52.5 (+37.2) | 27.1 (+13.2) | 48.1 (+22.1) |
| | MQA** | 83.4 | 84.7 | - | - | - | 73.2 | - |
| Flan-T5 XXL | Vanilla | 26.6 | 29.1 | 42.9 | 46.4 | 42.9 | 17.1 | 34.2 |
| | MQA | **50.6 (+24.0)** | **62.6 (+33.5)** | 53.5 (+10.6) | **61.6 (+15.2)** | **55.9 (+13.0)** | **53.3 (+36.2)** | **56.3 (+22.1)** |
| | MQA** | 83.9 | 86.0 | - | - | - | 86.6 | - |
| *InstructBLIP series (Zero-shot)* | | | | | | | | |
| Flan-T5 XL | Vanilla | 23.0 | 25.7 | 44.9 | 51.7 | 38.5 | 10.0 | 32.3 |
| | MQA | 39.7 (+16.7) | 49.9 (+24.2) | 53.3 (+8.4) | 59.0 (+7.3) | 52.9 (+14.4) | 22.7 (+12.7) | 46.3 (+14.0) |
| | MQA** | 81.2 | 80.9 | - | - | - | 70.3 | - |
| Flan-T5 XXL | Vanilla | 23.2 | 28.6 | 33.7 | 40.2 | 33.1 | 15.6 | 29.1 |
| | MQA | 48.1 (+24.9) | 58.8 (+30.2) | **55.2 (+21.5)** | 61.5 (+21.3) | 52.6 (+19.5) | 39.8 (+24.2) | 52.7 (+23.6) |
| | MQA** | 83.6 | 86.3 | - | - | - | 83.0 | - |
| Vicuna 7B | Vanilla | 9.8 | 17.8 | 18.5 | 24.9 | 13.2 | 2.7 | 14.5 |
| | MQA | 9.8 (+0.0) | 14.0 (-3.8) | 27.6 (+9.1) | 26.3 (+1.4) | 20.9 (+7.7) | 0.5 (-2.2) | 16.5 (+2.0) |
| | MQA** | 33.6 | 35.3 | - | - | - | 26.7 | - |
| Vicuna 13B | Vanilla | 14.0 | 15.0 | 17.5 | 26.1 | 8.1 | 1.4 | 13.7 |
| | MQA | 15.2 (+1.2) | 19.9 (+4.9) | 31.8 (+14.3) | 38.5 (+12.4) | 21.7 (+13.6) | 0.1 (-1.3) | 21.2 (+7.5) |
| | MQA** | 64.6 | 61.3 | - | - | - | 34.3 | - |

## 5 EXPERIMENTS

### 5.1 MAIN RESULTS

In zero-shot setting, our results derived from six MIE datasets are shown in Table 2, from which we make the following observations:

Firstly, across six datasets with six different models, MQA framework demonstrates significant performance improvements compared to vanilla method in 32 out of 36 evaluations, where most of them show an increase of over 10% in the F1 score. Particularly, the BLIP-2 series showcases substantial improvements, boasting an average increase of over 20% in F1 scores. Notably, the model built on Flan-5 XL demonstrates a remarkable increase of 37.2% in the F1 score on the MIED dataset, signifying an extraordinary 243% relative performance improvement compared to the vanilla method. These results corroborate the wide-ranging effectiveness of our proposed MQA framework.

Secondly, by employing MQA approach, LMMs notably outperform the most advanced large-scale language models (ChatGPT and GPT-4) with 175B parameters and beyond, across most datasets. This superiority becomes particularly evident when using the 10B parameter-level BLIP-2 with FlanT5 XXL on the MNER and MED tasks, where we demonstrate a significant performance advantage. Furthermore, our model consistently surpasses previous zero-shot SOTA models across various datasets and achieves comparable or better results than the 50-shot fine-tuning SOTA results on MNER. For comparison, our MQA's 50-shot fine-tuning results can be found in Section 5.2.

Thirdly, with either vanilla or MQA methods, the performance of InstructBLIP generally falls short compared to similarly scaled BLIP-2 models, despite that it is instruction-tuned with more data. We hypothesize that this disparity is due to the limited amount of task types for instruction tuning. Consequently, there are limited benefits when compared to its BLIP-2 upon receiving new instructions.

Table 3: Results of Flan-T5 XXL fine-tuned on 50 samples, with vanilla and MQA methods across six MIE datasets (%). We highlight F1 score improvements of MQA over vanilla in green.

| Methods | | MNER | | MRE | | MED | |
|---|---|---|---|---|---|---|---|
| | | Twitter15 | Twitter17 | MNRE-V1 | MNRE-V2 | Image | Text |
| 0-shot | Vanilla | 26.6 | 29.1 | 42.9 | 46.4 | 42.9 | 17.1 |
| | MQA | 50.6 (+24.0) | 62.6 (+33.5) | 53.5 (+10.6) | 61.6 (+15.2) | 54.2 (+11.3) | 53.6 (+36.5) |
| 50-shot | Vanilla | 28.9 | 31.5 | 45.1 | 48.0 | 45.2 | 16.8 |
| | MQA | 55.5 (+26.6) | 70.6 (+39.1) | 54.3 (+9.2) | 61.7 (+13.7) | 55.4 (+10.2) | 54.6 (+37.8) |

Fourthly, models based on Vicuna significantly underperform those based on Flan-T5, both with vanilla and MQA methods. This can be attributed to the fact that Vicuna is fine-tuned with open-ended conversations derived from user interactions with ChatGPT, instead of with traditional NLP tasks like Flan-T5, thereby restricting Vicuna's ability on MIE tasks.[2] That being said, our MQA still brings decent performance gains to Vicuna-based LMMs on average.

Lastly, we conduct an additional experiment by providing golden spans in MNER and MTED tasks (MQA** in Table 2). When golden spans are presented, the MQA results for all models demonstrate further enhancements. For instance, Flan-T5 XXL achieves F1 scores of 83.9%, 86.0%, and 86.6% on the Twitter15, Twitter17, and MTED datasets, respectively. The results, especially in Twitter15, are comparable with those of SOTA model (Chen & Feng, 2023) fine-tuned with full dataset, showing performance bottleneck of span extraction and the effectiveness of multi-choice QA per se.

## 5.2 FEW-SHOT FINE-TUNING RESULT

The 50-shot fine-tuning results are shown in Table 3. From these results, we can draw the following conclusions: (1) The performance of our MQA remains significantly superior compared to the vanilla method across the board. (2) The MQA method showcases improvements across all datasets and tasks. This is particularly noticeable in the MNER task, where it demonstrates a 4.9% and 8% increase in F1 score on the Twitter15 and Twitter17 datasets, respectively. Conversely, the model with vanilla method undergoes a slight performance degradation on the MTED task. (3) Following the 50-shot few-shot fine-tuning, our MQA method outperforms PGIM (Li et al., 2023a) on the Twitter15 dataset, which also undergoes 50-shot fine-tuning on Twitter, allowing our MQA method to achieve SOTA results across all six datasets.

## 5.3 EVALUATION OF MODEL ROBUSTNESS

Prompt robustness refers to the model's ability to accurately understand and respond to different prompts expressing the same task intent. Prior works (Lou et al., 2023; Sun et al., 2023) highlight that different instructions may lead to considerable performance fluctuations in language models. Therefore, to evaluate the instruction-following robustness of vanila and MQA methods, we manually rewrite four different instructions for each of them. In addition, Lu et al. (2022b) find the order of the input examples also has a significant impact on model performance. To verify the robustness of the input order of multi-choice QA, we randomly shuffle the answer options. All experiments are conducted using BLIP-2 model with Flan-T5 XXL language component.

**Evaluation of Robustness to Instruction Variants.** To evaluate the robustness of instruction-following, we select one representative dataset from each of the four tasks: MNER, MRE, MIED, and MTED. For each dataset, we manually write four diverse yet semantically equivalent instructions by rephrasing the initial instruction. Please refer to Appendix A.2 for details of these instructions for vanilla and MQA methods.

From the results in Table 4, we have the following observations: (1) Across various instructions on different tasks, MQA consistently outperforms vanilla prompting significantly. (2) MQA exhibits better overall robustness. To be more specific, in the MNER and MRE datasets, the model obtains a considerably low sample standard deviation of 0.1% and 0.4%, respectively, whereas vanilla, even under lower performance, shows a higher sample standard deviations of 6.1% and 3.8%. However,

---

[2]Vicuna is fine-tuned with data from https://sharegpt.com.

Table 4: Comparison of the robustness to instruction variants between vanilla and MQA methods (%). The final row presents the mean and sample standard deviation of the model performance under four instructions.

| Instruction | Variant | MNER-17 | | | MRE-v2 | | | MIED | | | MTED | | |
|---|---|---|---|---|---|---|---|---|---|---|---|---|---|
| | | P | R | F1 | P | R | F1 | P | R | F1 | P | R | F1 |
| Instruction1 | Vanilla | 20.0 | 53.7 | 29.1 | 33.8 | 74.1 | 46.4 | 37.6 | 49.9 | 42.9 | 9.7 | 69.4 | 17.1 |
| | MQA | 61.2 | 64.0 | 62.6 | 50.2 | 79.7 | 61.6 | 60.4 | 52.0 | 55.9 | 49.8 | 57.2 | 53.3 |
| Instruction2 | Vanilla | 15.9 | 52.2 | 24.4 | 29.7 | 62.7 | 40.3 | 30.3 | 58.3 | 39.8 | 9.6 | 73.6 | 17.0 |
| | MQA | 61.2 | 64.0 | 62.6 | 48.2 | 81.6 | 60.6 | 29.9 | 71.1 | 42.1 | 51.4 | 56.1 | 53.7 |
| Instruction3 | Vanilla | 11.5 | 50.7 | 18.8 | 31.0 | 66.9 | 42.3 | 28.4 | 66.9 | 39.9 | 9.3 | 71.1 | 16.5 |
| | MQA | 61.2 | 64.0 | 62.6 | 49.2 | 79.2 | 60.7 | 32.8 | 78.0 | 46.2 | 53.9 | 51.2 | 52.5 |
| Instruction4 | Vanilla | 9.1 | 47.9 | 15.2 | 29.4 | 61.4 | 37.5 | 28.3 | 67.5 | 39.9 | 9.4 | 72.0 | 16.5 |
| | MQA | 60.9 | 63.8 | 62.3 | 48.8 | 80.3 | 61.0 | 66.1 | 31.2 | 42.4 | 54.0 | 50.4 | 52.2 |
| $u \pm \sigma$ | Vanilla | | | $21.9 \pm 6.1$ | | | $41.6 \pm 3.8$ | | | $40.6 \pm 1.5$ | | | $16.8 \pm 0.3$ |
| | MQA | | | $62.5 \pm 0.1$ | | | $61.0 \pm 0.4$ | | | $46.6 \pm 6.4$ | | | $52.9 \pm 0.7$ |

Table 5: Robustness of our MQA method to input option order variants (%). The final row displays the mean and sample standard deviation of the performances across the four variants.

| Option Order | MNER-17 | | | MRE-v2 | | | MIED | | | MTED | | |
|---|---|---|---|---|---|---|---|---|---|---|---|---|
| | P | R | F1 | P | R | F1 | P | R | F1 | P | R | F1 |
| Order1 | 61.2 | 64.0 | 62.6 | 50.2 | 79.7 | 61.6 | 60.4 | 52.0 | 55.9 | 49.8 | 57.2 | 53.3 |
| Order2 | 63.4 | 62.8 | 63.1 | 49.6 | 80.2 | 61.3 | 55.9 | 53.5 | 54.7 | 47.7 | 52.1 | 49.8 |
| Order3 | 60.4 | 63.7 | 62.0 | 49.8 | 79.8 | 61.3 | 57.9 | 53.0 | 55.3 | 49.2 | 55.1 | 52.0 |
| Order4 | 60.4 | 64.0 | 62.1 | 50.2 | 80.5 | 61.9 | 57.3 | 55.4 | 56.3 | 48.8 | 52.5 | 50.6 |
| $u \pm \sigma$ | | | $62.5 \pm 0.5$ | | | $61.5 \pm 0.3$ | | | $55.5 \pm 0.7$ | | | $51.4 \pm 1.5$ |

an exception is observed in the MIED dataset wherein our model exhibited a relatively high variation. We hypothesize this might be due to the fact that MIED task solely relies on images and BLIP-2 series models have not been exposed to image classification prompts during pre-training.

**Evaluation of Robustness to Input Order.** To evaluate our MQA framework's robustness to varying input orders, we design four distinct input formats by permuting the arrangement of multi-choice options in a random manner. For comprehensiveness, we also evaluate the robustness on four datasets from four MIE tasks with results shown in Table 5. We find that our MQA exhibits a high degree of robustness to varying input orders. The performance shows little fluctuation, demonstrating the model's effectiveness in handling changes in the order of multi-choice options. Notably, our approach achieves less than 1% sample standard deviation on the MNER, MRE, and MIED datasets. These results confirm the MQA framework's high robustness to variations in input order.

The robustness to instruction-following and input order of MQA framework indicates less prompt engineering effort and high practicalness in the real-world scenarios.

## 6 CONCLUSION

In this study, we present a novel MQA framework, aiming to unify three diverse MIE tasks across various settings, effectively addressing concerns related to task generalization and data scarcity. Upon six off-the-shelf LMMs, MQA consistently achieves significant performance improvements compared to vanilla prompting strategies. Remarkably, without necessitating any fine-tuning, the MQA framework surpasses prior task-specific low-resource SOTA methods across six datasets and achieves comparable or even better results than much larger language models such as ChatGPT and GPT-4 on most of the datasets. In addition, MQA demonstrates robustness to various instructions and input orders, and exhibits effectiveness in the few-shot setting. All these desirable properties indicate that the proposed MQA framework is a powerful and unified tool for MIE tasks. More importantly, our experiments suggest that, aligning diverse downstream multimodal task formats (e.g., MIE) with the task forms that LMMs have been pre-trained on (e.g., VQA), holds a promising direction for enhancing the utility of LMMs in downstream tasks.

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

# A APPENDIX

## A.1 PROMPTS UTILIZED FOR VANILLA AND MQA APPROACHES

The specific prompts employed in the Vanilla and MQA approaches for each task are presented in Table. 6 (MNER), Table. 7 (MIED), Table. 8 (MRE), Table. 9 (MTED).

## A.2 INSTRUCTIONS VARIANTS WITHIN THE PROMPT USED FOR THE ROBUSTNESS EVALUATION OF VANILLA AND MQA

In the robustness evaluation experiment, various instruction variants within the prompts designed for vanilla can be observed in Table 10 (MNER), Table 11 (MIED), Table 12 (MRE), and Table 13 (MTED). Meanwhile, instruction variants for the MQA during the multi-choice QA stage are presented in Table 14 (MNER), Table 15 (MIED), Table 16 (MRE), and Table 17 (MTED).

## A.3 PROMPTS FOR CHATGPT/GPT4

We leverage the OpenAI API to execute the calls, utilizing the model codenamed *gpt3.5-turbo* for ChatGPT, and *gpt4* for GPT-4. The prompts used to assess the performance of ChatGPT/GPT-4 on text-based MIE tasks are displayed in Table 18.

Table 6: Prompt formats used in Vanilla and MQA approaches for MNER task.

| Formulations | Prompts |
|---|---|
| Vanilla | Please choose the entity spans corresponding to the [Entity category $E_c$] entity type that can be inferred from the given sentence.

Entities: |
| MQA - Span Extraction | Please list all named entity mentions in the sentence that fits the [Entity category $E_c$] category. Answer format is entity1, entity2, entity3.

Sentence: [Sentence $S$]

Answer: |
| MQA - Multi-choice QA | Choose the right answer about the entity that can be inferred from the given sentence.

Sentence: [Sentence $S$]

Options:
A. [Predicted Entity Span $E_{ps}$] is a location entity
B. [Predicted Entity Span $E_{ps}$] is a person entity
C. [Predicted Entity Span $E_{ps}$] is an organization entity
D. [Predicted Entity Span $E_{ps}$] is not a named entity or does not belong to type [location, person, organization, miscellaneous]

Which answer can be inferred from the given Sentence?
Answer: |

Table 7: Prompt formats used in Vanilla and MQA approaches for MIED task.

| Formulations | Prompts |
|---|---|
| Vanilla | Given an image and a sentence, classify which type of activity can be inferred.

Sentence: This is an image attached to a news article.

All possible types are listed below:

- Movement:Transport
- Contact:Phone-Write
- Conflict:Attack
- Contact:Meet
- Justice:Arrest-Jail
- Conflict:Demonstrate
- Life:Die
- Transaction:Transfer-Money
- None

Type: |
| MQA - Multi-choice QA | Given an image, classify which type of activity can be inferred.

Sentence: This is an image attached to a news article.

All possible types are listed below:

Options:
A. The image with the sentence describes the Transportation of Movement event
B. The image with the sentence describes the Phone or Write of Contact event
C. The image with the sentence describes the Attack with no death of Conflict event
D. The image with the sentence describes the Meet of Contact event
E. The image with the sentence describes the Arrest of Criminal event
F. The image with the sentence describes the Demonstration of Conflict event
G. The image with the sentence describes the Death of Life event
H. The image with the sentence describes the Money Transfer of Transaction event
I. The image describes no event

Which answer can be inferred from the image?
Answer: |

Table 8: Prompt formats used in Vanilla and MQA approaches for MRE task. We adopt the relation template in MQA - Multi-choice QA following Zhang et al. (2023a)

| Formulations | Prompts |
|---|---|
| Vanilla | Given the image and the text, select the relation between Entity 1 and Entity 2 that can be inferred from the given sentence. The image may provide fine-grained information about the entities. All possible relations are listed below:

-[Possible Relation 1]
-[Possible Relation 2]
-None

Sentence: [Sentence $S$]
Relation: |
| MQA - Multi-choice QA | In light of the provided sentence, determine which option is the most feasible inference. The image may present detailed information about the entities.

Sentence: [Sentence $S$]

Options:
A. [Entities in Relation 1 Template]
B. [Entities in Relation 2 Template]
C. [Entities in NoTA Relation Template]

Which option is the most possible inference?
Option: |

Table 9: Prompt formats used in Vanilla and MQA approaches for MTED task.

| Formulations | Prompts |
|---|---|
| Vanilla | Given an image and a sentence, determine which word can infer the [Event category $E_c$] activity.

Sentence: [Sentence $S$]

Words: |
| MQA - Span Extraction (Pre-process) | Determine which option can be inferred from the given sentence.

Sentence: [Sentence $S$]

Which option can be inferred from the given Sentence?

Options:
A. Activities involving the movement or transportation of people or goods from one place to another
B. Interactions between individuals through phone calls or written communication
C. Aggressive actions or assaults by one party against another
D. Instances where individuals physically meet or come into contact with each other
E. Incidents involving the arrest and subsequent detention in jail or custody of individuals
F. Public displays of disagreement or protest to express opinions or demands
G. The life of a person ends
H. The exchange of money or financial resources between parties |
| MQA - Span Extraction | Please choose the most possible trigger word from the verbs and nouns in the sentence that reflect the [Predicted Event category $E_{pc}$] event. Note that trigger words can only be noun or verb. Answer format is word1

Sentence: [Sentence $S$]

Answer: |
| MQA - Multi-choice QA | Determine which option can be inferred from the given sentence and image.

Sentence: [Sentence $S$]

Answer candidates:
A. The word [Predicted Trigger Word $T_p$] is a common word and does not reflect any of the other event
B. The word [Predicted Trigger Word $T_p$] is the key of the Transport action, which is a subtype of Movement event
C. The word [Predicted Trigger Word $T_p$] is the key of the PhoneWrite action, which is a subtype of Contact event
D. The word [Predicted Trigger Word $T_p$] is the key of the conflict but no death action, which is a subtype of Conflict event
E. The word [Predicted Trigger Word $T_p$] is the key of the Meeting action, which is a subtype of Contact event
F. The word [Predicted Trigger Word $T_p$] is the key of the Crime Arrest or sent into Jail action, which is a subtype of Justice event
G. The word [Predicted Trigger Word $T_p$] is the key of the Demonstrate action, which is a subtype of Conflict event
H. The word [Predicted Trigger Word $T_p$] is the key of the Die action, which is a subtype of Life event
I. The word [Predicted Trigger Word $T_p$] is the key of the Transfer-Money action, which is a subtype of Transaction event

Which answer can be inferred?

Answer: |

Table 10: Instruction variants in prompts used in the Vanilla approach for robustness evaluation in MNER task. Variations in the instructions are highlighted in pink.

| Formulations | Prompts |
|---|---|
| Instruction 1 | Please choose the entity spans corresponding to the [Entity category $E_c$] entity type that can be inferred from the given sentence.

Entities: |
| Instruction 2 | Choose the spans of [Entity category $E_c$] entity that can be inferred from the given sentence.

Entities: |
| Instruction 3 | Please choose the spans related to the [Entity category $E_c$] category that can be deduced from the presented sentence.

Entities: |
| Instruction 4 | Decide on the spans associated with the [Entity category $E_c$] entity category that can be inferred from the given sentence.

Entities: |

Table 11: Instruction variants in prompts used in the Vanilla approach for robustness evaluation in MIED task. Variations in the instructions are highlighted in pink. The content of possible types of events are identical across several instruction variants, thus they are abbreviated with "......". Please refer to Table 7 for a detailed overview.

| Formulations | Prompts |
|---|---|
| Instruction 1 | Given an image and a sentence, classify which type of activity can be inferred.

Sentence: This is an image attached to an news article.

All possible types are listed below:
......

Type: |
| Instruction 2 | Given an image, classify which type of activity can be inferred.

Sentence: This is an image attached to a news article.

All possible types are listed below:
......

Type: |
| Instruction 3 | Given an image, identify the event-related type that can be deduced from the provided image.

Sentence: This is an image attached to an news article.

All possible types are listed below:
......

Type: |
| Instruction 3 | Given the image, please choose which type of activity can be inferred.

Sentence: This is an image attached to a news article.

All possible types are listed below:
......

Type: |

Table 12: Instruction variants in prompts used in the Vanilla approach for robustness evaluation in MRE task. Variations in the instructions are highlighted in pink.

| Formulations | Prompts |
|---|---|
| Instruction 1 | Given the image and the text, select the relation between Entity 1 and Entity 2 that can be inferred from the given sentence, The image may provide fine-grained information about the entities. All possible relations are listed below:

-[Possible Relation 1]
-[Possible Relation 2]
-None

Sentence: [Sentence $S$]
Relation: |
| Instruction 2 | Given the image and the text, determine which relation between Entity 1 and Entity 2 can be deduced. The image might offer detailed insights about the entities. All possible relations are listed below:

-[Possible Relation 1]
-[Possible Relation 2]
-None

Sentence: [Sentence $S$]
Relation: |
| Instruction 3 | Based on the image and the text, determine which relation is the most possible between Entity 1 and Entity 2. The image could offer intricate details about the entities. All possible relations are listed below:

-[Possible Relation 1]
-[Possible Relation 2]
-None

Sentence: [Sentence $S$]
Relation: |
| Instruction 4 | In light of the image and the text, determine what relation between Entity 1 and Entity 2 is the most feasible. The image may present detailed information about the entities. All possible relations are listed below:

-[Possible Relation 1]
-[Possible Relation 2]
-None

Sentence: [Sentence $S$]
Relation: |

Table 13: Instruction variants in prompts used in the Vanilla approach for robustness evaluation in MTED task. Variations in the instructions are highlighted in pink.

| Formulations | Prompts |
|---|---|
| Instruction 1 | Given an image and a sentence, determine which word can infer the [Event category $E_c$] activity.

Sentence: [Sentence $S$]

Words: |
| Instruction 2 | Given an image and a sentence, classify the right word in the sentence that can infer the [Event category $E_c$] activity.

Sentence: [Sentence $S$]

Words: |
| Instruction 3 | Given an image and a sentence, determine the right word in the sentence that can infer the [Event category $E_c$] event.

Sentence: [Sentence $S$]

Words: |
| Instruction 4 | Given an image and a sentence, select the correct word within the sentence that can infer the [Event category $E_c$] activity.

Sentence: [Sentence $S$]

Words: |

Table 14: Instruction variants in prompts used in the Multi-choice QA stage of MQA approach for robustness evaluation in MNER task. Variations in the instructions are highlighted in pink.

| Formulations | Prompts |
|---|---|
| Instruction 1 | Choose the right answer about the entity that can be inferred from the given sentence.

Sentence: [Sentence $S$]

Options:
A. [Entity Span] is a location entity
B. [Entity Span] is a person entity
C. [Entity Span] is an organization entity
D. [Entity Span] is not a named entity or does not belong to type [location, person, organization, miscellaneous]

Which answer can be inferred from the given sentence?
Answer: |
| Instruction 2 | Select the correct option regarding the entity that can be deduced from the provided sentence.

Sentence: [Sentence $S$]

Options:
A. [Entity Span] is a location entity
B. [Entity Span] is a person entity
C. [Entity Span] is an organization entity
D. [Entity Span] is not a named entity or does not belong to type [location, person, organization, miscellaneous]

Which answer can be inferred from the given sentence?
Answer: |
| Instruction 3 | Pick the accurate answer concerning the entity that can be implied from the presented sentence.

Sentence: [Sentence $S$]

Options:
A. [Entity Span] is a location entity
B. [Entity Span] is a person entity
C. [Entity Span] is an organization entity
D. [Entity Span] is not a named entity or does not belong to type [location, person, organization, miscellaneous]

Which answer can be inferred from the given sentence?
Answer: |
| Instruction 4 | Choose the right answer pertaining to the entity that can be concluded from the given sentence.

Sentence: [Sentence $S$]

Options:
A. [Entity Span] is a location entity
B. [Entity Span] is a person entity
C. [Entity Span] is an organization entity
D. [Entity Span] is not a named entity or does not belong to type [location, person, organization, miscellaneous]

Which answer can be concluded from the given sentence?
Answer: |

Table 15: Instruction variants in prompts used in the Multi-choice QA stage of MQA approach for robustness evaluation in MIED task. Variations in the instructions are highlighted in pink. The options are identical across several instruction variants, thus they are abbreviated with "......". Please refer to Table 7 for a detailed overview.

| Formulations | Prompts |
|---|---|
| Instruction 1 | Given an image, classify which type of activity can be inferred. 

 Sentence: This is an image attached to an news article. 

 Options: 
 ...... 

 Which answer can be inferred from the image? 
 Answer: |
| Instruction 2 | Determine which choice of activity is relevant to the image. 

 Options: 
 ...... 

 Which answer is relevant to the image? 
 Answer: |
| Instruction 3 | Please determine which choice of activity can be inferred from the image. 

 Options: 
 ...... 

 Which option can be inferred from the image? 
 Answer: |
| Instruction 4 | Identify the option that can be deduced from the picture. 

 Sentence: This is an image attached to an news article. 

 Options: 
 ...... 

 What can be deduced from the picture? 
 Answer: |

Table 16: Instruction variants in prompts used in the Multi-choice QA stage of MQA approach for robustness evaluation in MRE task. Variations in the instructions are highlighted in pink.

| Formulations | Prompts |
|---|---|
| Instruction 1 | In light of the provided sentence, determine which option is the most feasible inference. The image may present detailed information about the entities.

Sentence: [Sentence $S$]

Options:
A. [Entities in Relation 1 Template]
B. [Entities in Relation 2 Template]
C. [Entities in NoTA Relation Template]

Which option is the most possible inference?
Option: |
| Instruction 2 | Select the option that can be inferred from the given sentence. The image may provide fine-grained information about the entities.

Sentence: [Sentence $S$]

Options:
A. [Entities in Relation 1 Template]
B. [Entities in Relation 2 Template]
C. [Entities in NoTA Relation Template]

Which option can be inferred from the given sentence and image?
Option: |
| Instruction 3 | From the provided sentence, determine which option can be deduced. The image might offer detailed insights about the entities.

Sentence: [Sentence $S$]

Options:
A. [Entities in Relation 1 Template]
B. [Entities in Relation 2 Template]
C. [Entities in NoTA Relation Template]

Which option can be deduced from the given sentence and image?
Option: |
| Instruction 4 | Based on the given sentence, determine which option is the most possible deduction. The image could offer intricate details about the entities.

Sentence: [Sentence $S$]

Options:
A. [Entities in Relation 1 Template]
B. [Entities in Relation 2 Template]
C. [Entities in NoTA Relation Template]

Which option is most possible?
Option: |

Table 17: Instruction variants in prompts used in the Multi-choice QA stage of MQA approach for robustness evaluation in MTED task. The answer candidates are identical across several instruction variants, thus they are abbreviated with "......". Please refer to Table 9 for a detailed overview. Variations in the instructions are highlighted in pink.

| Formulations | Prompts |
|---|---|
| Instruction 1 | Determine which option can be inferred from the given sentence and image.

Sentence: [Sentence $S$]

Answer candidates:
......

Which answer can be inferred?

Answer: |
| Instruction 2 | Identify the choice that can be deduced from the provided sentence and image.

Sentence: [Sentence $S$]

Answer candidates:
......

Which answer can be inferred?

Answer: |
| Instruction 3 | Choose the right answer about a word in the sentence that can be inferred.

Sentence: [Sentence $S$]

Answer candidates:
......

Which answer can be inferred?

Answer: |
| Instruction 4 | Select the correct answer concerning a word within the sentence that can be inferred.

Sentence: [Sentence $S$]

Answer candidates:
......

Which answer can be inferred?

Answer: |

Table 18: Prompts employed for ChatGPT/GPT4 in the context of text-based MIE tasks.

| Task | Prompts |
|------|---------|
| MNER | Extract the entity span that belongs to [Entity category $E_c$] entity category from the provided sentence. Output format is entity1. If no [Entity category $E_c$] entity can be inferred, respond with "". 

 Output: |
| MRE | Determine the what relation between Entity 1 and Entity 2 is according to the text. All possible relations are listed below: 

 -[Possible Relation 1] 
 -[Possible Relation 2] 
 -None 

 Sentence: [Sentence $S$] 
 Relation: |
| MTED | Extract the most one trigger word from the provided sentence that infer the [Event category $E_c$] activity. Note that the trigger word can only be a noun or verb. The answer format should be word1. If no trigger word reflecting the [Event category $E_c$] activity can be identified, respond with "". 

 Sentence: [Sentence $S$] 

 Words: |

