# OpenReview forum: "Multimodal Question Answering for Unified Information Extraction"
_ICLR.cc/2024/Conference — ICLR 2024 Conference Withdrawn Submission_

### Official Review · Reviewer_F9v2 · 2023-10-26

**Soundness:** 3 good
**Presentation:** 3 good
**Contribution:** 2 fair
**Rating:** 3
**Confidence:** 4

**Summary:**

The paper proposes a Multimodal Question Answering (MQA) framework that addresses the limitations of existing Multimodal Information Extraction (MIE) models.
The MQA framework unifies three MIE tasks into a single pipeline which specifically involves 1) span extraction and 2) multi-choice QA, improving efficacy and generalization across multiple datasets.
The authors conduct experiments on three MIE benchmark datasets, some discussions are provided, such as the evaluation of model robustness.

**Strengths:**

1. The paper proposes to unify three Multimodal Information Extraction (MIE) tasks (e.g., multimodal named entity extraction, multimodal relation extraction, and multimodal event detection) into a single pipeline, offering a more efficient and generalizable solution.
2. The MQA framework outperforms baseline models in terms of efficacy and generalization across multiple datasets.

**Weaknesses:**

1. The running title might be problematic or misleading. There seems to be a disconnect between the title and the content of the paper. While the title underscores the theme of unified information extraction, the body of the paper leans more towards the unification of different modalities in information extraction. Furthermore, the current approach brings together just three MIE tasks, leaving out others like multimodal aspect term extraction [1] and multimodal opinion extraction [2], which are also the very common MIE tasks. To truly capture the essence of a comprehensive unified MIE framework, these tasks should be integrated as well.

2. The paper overlooks some pivotal and recent works related to MIE, such as [3-4], which could have added depth to the discussion and provided a broader context.


3. I harbor reservations regarding the distinctiveness of the technical novelty. Translating (text-based) information extraction tasks into a question-answering (QA) or machine reading comprehension (MRC) framework has become commonplace in the NLP community [5-11], having undergone extensive scrutiny over time. Moreover, despite the broad exploration of QA-centric methodologies for information extraction, the authors fall short in contrasting their novel methods with existing QA-driven frameworks.


4. The feasibility of employing ChatGPT for multimodal information extraction remains uncertain, given its inherent design to primarily process text. This limitation demands clarification.

5. There are observable typos or writing mistakes in the paper, for example in Table 2: "the figures 61.7 derived from GPT-4 are denoted as the best results on MERE-V1,  similar to 52.2". Further careful proofreading is needed.

6. The paper lacks some crucial in-depth analyses that would elucidate the underlying mechanisms behind the proposed methods. Such insights, including comparisons between large language models (LLM) and supervised models, would be invaluable for readers. Merely showing shallow numerical comparisons is never enough.



[1] A Multi-modal Approach to Fine-grained Opinion Mining on Video Reviews

[2] Joint Multi-modal Aspect-Sentiment Analysis with Auxiliary Cross-modal Relation Detection

[3] Multimodal Relation Extraction with Cross-Modal Retrieval and Synthesis

[4] Information Screening whilst Exploiting! Multimodal Relation Extraction with Feature Denoising and Multimodal Topic Modeling

[5] QuAChIE: Question Answering based Chinese Information Extraction System

[6] QA4IE: A Question Answering Based Framework for Information Extraction

[7] A Multi-turn Machine Reading Comprehension Framework with Rethink Mechanism for Emotion-Cause Pair Extraction

[8] Enhanced Machine Reading Comprehension Method for Aspect Sentiment Quadruplet Extraction

[9] Have my arguments been replied to? Argument Pair Extraction as Machine Reading Comprehension.

[10] A New Entity Extraction Method Based on Machine Reading Comprehension

[11] NER-MQMRC: Formulating Named Entity Recognition as Multi Question Machine Reading Comprehension

**Questions:**

Given the widespread adoption of QA frameworks in the NLP domain, how does the proposed MQA approach distinguish itself, particularly in terms of technical innovations?

---

### Official Review · Reviewer_HQKu · 2023-10-30

**Soundness:** 2 fair
**Presentation:** 2 fair
**Contribution:** 1 poor
**Rating:** 3
**Confidence:** 5

**Summary:**

This paper proposes a multimodal question-answering framework, namely MQA, to tackle three multimodal information extraction (MIE) tasks, including multimodal named entity recognition (MNER), multimodal relation extraction (MRE), and multimodal event detection (MED). Specifically, based on the instruction-following of large multimodal models (LMMs), all of the three tasks are reformulated by the multiple-choice QA template to respectively classify entity type, relation type, and event type. Besides, before classification, MNER and text-centric MED (MTED) tasks also employ type-specific span-extraction QA templates to attain candidate spans. Experimentally, without any fine-tuning, the MQA framework improves the zero-shot performance of two LMMs, i.e. BLIP-2 and InstructBLIP, outperforming two more advanced models, i.e. ChatGPT and GPT4, with vanilla reformulation format. When fine-tuned on few-shot samples, the performance can be further improved. Experiments also demonstrate that MQA can improve the robustness of LMMs to instruction variants and order of input sequence.

**Strengths:**

This paper makes a simple yet somehow effective attempt to unify various MIE tasks.
This paper has performed relatively extensive experiments under the setting of different MIE tasks and LMM scales.
This paper is well-written and quite easy to follow.

**Weaknesses:**

The main weaknesses are three-fold:
Overall, this paper lacks novelty and makes limited contributions. QA-based reformulation (both span-based and multiple-choice) is one of the most typical and long-standing formats for unifying various IE tasks in the NLP community, especially in the era of large-scale models [1-4]. Although this paper targets MIE and incorporates another modality, i.e. an input image paired with the text, it does not pay more attention to understanding the image content and interactions between the two modalities. Therefore, this paper utilizes such a QA format for unified MIE, which contributes less from the perspective of technique. Furthermore, this paper slightly overclaims the contribution in Section 1, i.e. it is a stretch to claim to unify MIE tasks as there are only three evaluated MIE tasks in the experiment setting.

Secondly, the motivation and usage of QA-based reformulation are problematic and unconvincing. Specifically, with the assistance of LMM, a straightforward and intuitive way to reformulate various tasks is to prompt the LMM with task description and optional in-context exemplars, which is widely used in vison-language instruction-following [5], and dubbed as vanilla prompt in this paper (Section 4.3.1). However, this paper leverages two kinds of QA-based reformulation templates, i.e. multi-choice QA template and span-extraction QA template. The reason and motivation behind this are not well explained (Section 1). Furthermore, the former is used for all three MIE tasks to output the entity type, relation type, or event type, while the latter is only engaged in MNER and MTED to obtain better candidate spans. This makes it not very clear whether the good performance can be attributed to the accurate span localization rather than the unified multi-choice QA template. This issue is also portrayed in Table 2 to some extent. For example, without the span localization, the MRE task seems to achieve relatively less gain compared to MNER and MTED tasks.

Last, in response to the above two points, the author may argue that MQA exceeds SOTA and large-scale model baselines substantially, especially when not extracting candidate spans and only using multiple-choice QA templates. However, in that case, the third concern comes to the foreground, i.e. more comparison and ablation experiments are required to justify the effectiveness of MQA and multi-choice QA template. On the one hand, more advanced LMM [6], especially those that have been instruction-tuned in this kind of multi-choice QA template should be discussed and had better be experimentally compared if possible. In addition, as stated by the authors (Section 4.4), ChatGPT and GPT-4 are used to conduct comparisons by only accepting the text input, and they even achieve comparable results to MQA in the MRE task (Table 2). Although they have a larger scale than examined LMM backbones, they completely miss a modality and this comparison makes no sense. On the other hand, this paper does not demonstrate the respective contribution of the multiple-choice QA template and span-extraction QA template, especially on the MNER and MTED tasks.

[1] UniEX: An Effective and Efficient Framework for Unified Information Extraction via a Span-extractive Perspective, ACL 2023
[2] Unified Structure Generation for Universal Information Extraction, ACL 2022
[3] Exploring the limits of transfer learning with a unified text-to-text transformer, JMLR 2020
[4] The natural language decathlon: Multitask learning as question answering, arXiv 2018
[5] VisIT-Bench: A Benchmark for Vision-Language Instruction Following Inspired by Real-World Use, arXiv 2023
[6] MME: A Comprehensive Evaluation Benchmark for Multimodal Large Language Models, arXiv 2023

**Questions:**

1. Is a span-extraction QA prompt necessary for MNER and MTED, and is it possible to use a span-extraction QA prompt to extract span for MRE trigger words?
2. What is the performance when only using multiple-choice QA prompts?
3. What is the difference between the answer option construction (Section 3.4) of multiple-choice QA prompt and the verbalizer of prompt-tuning?
4. What is the objective when conducting 50-shot fine-tuning for backbone LMMs, and what is the number of trainable parameters during this process?

---

### Official Review · Reviewer_JWLd · 2023-10-30

**Soundness:** 4 excellent
**Presentation:** 3 good
**Contribution:** 4 excellent
**Rating:** 8
**Confidence:** 4

**Summary:**

The paper introduces MQA, a novel and unified framework for multimodal information extraction tasks, encompassing tasks such as multimodal named entity recognition (MNER), multimodal relation extraction (MRE), and multimodal event detection (MED). Unlike traditional approaches that require separate prompts for each task when using LLMs, MQA unifies these tasks by transforming them into a multiple-choice question-answering format with a standardized template. For tasks like MNER that can't be directly converted, an intermediate step for span extraction is introduced prior to the QA phase.

The study conducts a thorough set of experiments on a total of six datasets, benchmarking the results against robust baselines including SOTA models, recent LLMs, and latest GPT models. The results consistently demonstrate that the proposed framework outperforms the baselines by a significant margin, both in zero-shot and few-shot scenarios. Additionally, the study showcases how MQA enhances stability across different instruction variations.

**Strengths:**

1. The proposed MQA achieves SOTA results on six datasets across three MIE subtasks, showcasing significant advancements.
2. The MQA framework exhibits impressive generalization and wide applicability. It effectively integrates with various LLMs, consistently outperforming their vanilla versions. Additionally, during robustness testing, MQA displays relatively small performance variation under different prompting strategies and input orders, underscoring its robustness and adaptability.
3. The proposed method is straightforward and tidy. For specific subtasks, it merely necessitates an input reformatting.

Overall, I think the work will benefits a lot to researchers and the community.

**Weaknesses:**

1. To unify MNER, MRE, and MED tasks, an additional span extraction is introduced for some tasks like MNER & MTED, which adds extra complexity to the overall system.

**Questions:**

In Section 3.1 - Multimodal Relation Extraction section paragraph,  should the tuple be (T, I, E_h, E_t) ?

In Table 6, is it a typo for the missing "Sentence: [Sentence S]"?

What is the "pre-process" step in table 9?

---

### Official Review · Reviewer_jjah · 2023-11-13

**Soundness:** 2 fair
**Presentation:** 3 good
**Contribution:** 2 fair
**Rating:** 5
**Confidence:** 3

**Summary:**

The paper introduces the Multimodal Question Answering (MQA) framework, a novel approach of unifying Multimodal Information Extraction (MIE) tasks, including Multimodal Named Entity Recognition (MNER), Multimodal Relation Extraction (MRE), and Multimodal Event Detection (MED). MQA innovatively reformulates these tasks into a multiple-choice question-answering format, with an additional span extraction step for certain tasks. Extensive experiments across six datasets demonstrate MQA's superior performance over state-of-the-art models, including large multimodal models (LMMs) and recent versions of GPT. The framework shows notable advancements in both zero-shot and few-shot scenarios and exhibits robustness against various instruction variations.

**Strengths:**

1. The framework demonstrates impressive generalization capabilities and stability, outperforming traditional models and LMMs, including ChatGPT and GPT-4.
2. MQA effectively enhances the performance of LMMs in MIE tasks.
3. The method's straightforward nature, requiring only input reformatting for specific subtasks, is commendable.

**Weaknesses:**

1. The addition of an extra span extraction step for certain tasks like MNER adds complexity. Also, the framework focuses on only three MIE tasks, not covering the full range of multimodal tasks.
2. The QA-based reformulation method, though effective, is not a new concept in NLP. Moreover, the paper does not delve deeply into understanding the interactions between different modalities.
3. The technical novelty of MQA in comparison to existing QA-driven frameworks in NLP is not sufficiently highlighted.

**Questions:**

1. Is the span extraction step essential for tasks like MNER and MTED, and could it be applied to MRE trigger words?
2. What are the results when employing solely the multiple-choice QA prompts, without span extraction?
3. How does MQA technically distinguish itself from existing QA frameworks in NLP?